# Combined Application of Myo-Inositol and Corn Steep Liquor from Agricultural Waste Alleviate Salt Stress in *Brassica rapa*

**DOI:** 10.3390/plants12244110

**Published:** 2023-12-08

**Authors:** Xinjun Zhang, Xian Wang, Wenna Zhang, Qing Chen

**Affiliations:** 1Beijing Key Laboratory of Farmyard Soil Pollution Prevention-Control and Remediation, College of Resources and Environmental Sciences, China Agricultural University, Beijing 100193, China; zhang0902@cau.edu.cn (X.Z.); wx1996ll@163.com (X.W.); qchen@cau.edu.cn (Q.C.); 2Beijing Key Laboratory of Growth and Developmental Regulation for Protected Vegetable Crops, China Agricultural University, Beijing 100193, China

**Keywords:** salt stress, cabbage, corn steep liquor, myo-inositol

## Abstract

Salinity poses a significant threat to plant growth through induction of osmotic and ionic stress and disruption of nutrient absorption. Biostimulants derived from agricultural waste offer a sustainable solution to alleviate salt-induced damage to plants and contribute to a circular and sustainable economy. In this study, we applied a combination of myo-inositol and corn steep liquor from waste sources to seedling cabbage (*Brassica rapa* subsp. *pekinensis*) and investigated their effects on plant growth under NaCl-simulated salt stress. Different concentrations of myo-inositol and corn steep liquor were applied to the roots, revealing that 150 mM NaCl significantly inhibited the growth and physiological metabolism of cabbage seedlings. Substrate application of myo-inositol, corn steep liquor, and their combination materials increased biomass, photosynthetic pigments, soluble sugars, soluble proteins, and the contents of K^+^, Ca^2+^, and Mg^2+^ in cabbage under salt stress conditions, while reducing malondialdehyde, electrolyte leakage, Na+ content, and the ratios of Na^+^/K^+^, Na^+^/Ca^2+^, and Na^+^/Mg^2+^. Therefore, root application of myo-inositol, corn steep liquor, and myo-inositol–corn steep liquor combination materials enhanced photosynthesis and enhanced cabbage salt stress resistance by maintaining cell osmotic and ion balance. The most pronounced positive effects were observed in the treatment with 0.1 mL L^−1^ corn steep liquor +288 mg L^−1^ myo-inositol. This study provides a theoretical basis and technical guidance for the combined utilization of myo-inositol and corn steep liquor to boost early growth and salt resistance in crops.

## 1. Introduction

Salinity is one of the most serious environmental constraints on agricultural productivity, particularly in arid and semi-arid regions. Soil salt ions, such as Na^+^ and Cl^−^, restrict roots from the absorption of water and nutrients, resulting in osmotic stress and ion toxicity in crops [1]. The influx of deleterious salt ions into plant cells induces toxicity, leading to metabolic disorders, disruption of enzymes and macromolecules, and structural damage to organelles and membranes. Such adverse effects extend to inhibiting vital biological processes like respiration, photosynthesis, and protein synthesis [2]. Therefore, salt stress hampers plant growth and ultimately reduces crop yield and quality by disrupting various physiological and molecular activities within the plants [3].

Napa cabbage (*Brassica rapa* subsp. *pekinensis*), one of the most widely cultivated vegetables in China, holds significant economic importance as a key crop in the country [4]. Nevertheless, the widespread cultivation of cabbage faces challenges from salt stress, disrupting not only the normal growth of cabbage but also affecting the overall development of the vegetable economy [5]. Salt stress induced a significant reduction in photosynthetic pigment level and stomatal conductance, resulting in compromised photosynthesis [6]. Prolonged exposure to salt stress triggers the accumulation of reactive oxygen species (ROS) in the cabbage, leading to cellular lipid peroxidation [7]. In addition, salt stress results in a substantial influx of Na^+^ into cabbage cells, disrupting ion balance and causing ion toxicity [8].

Agricultural wastes often contain bioactive substances, so the development of plant growth stimulants from agricultural wastes can play a positive role in promoting plant growth and resilience [9]. Corn steep liquor (CSL), a concentrated byproduct from corn starch production, is an acidic viscous liquid rich in various nutrients, including amino acids, sugars, proteins, vitamins, and minerals [10]. The diverse nutritional composition of CSL allows for versatile applications across various fields, such as additives in biotechnological production media, animal feed, surfactant production, and a component in biostimulants [11]. The presence of bioactive molecules such as organic acids and plant growth regulators in CSL provides a basis for its use in soil improvement and biostimulant research [12]. Phenolic compounds in CSL exhibit inhibitory effects on oxidative stress, indicating that CSL is a useful source of phenolic antioxidants [13]. Treating soybeans with 1% CSL accelerated nutrient transport and uptake, thereby promoting the germination, growth, and early maturity of soybeans [14]. CSL has been found to enhance plant nutrient utilization by fostering the growth of nitrogen-fixing and phosphate-solubilizing bacteria [15], which supports the beneficial effect of CSL on lettuce root microbiota [16]. Utilization of a mixture of CSL and other organic materials significantly increased maize yield via increasing soil organic carbon content [17]. In summary, CSL can optimize soil structure, promote crop growth and development, and improve plant performance under abiotic stress. Therefore, it can be expected that the application of CSL in agriculture can not only help consume huge amounts of waste byproducts but also reduce the dependence on chemical fertilizers and improve crop health.

Under salt stress, plants synthesize and accumulate osmotic active substances, such as proline, to maintain the normal function of enzymes and ensure membrane integrity by increasing cell osmotic pressure [18]. Myo-inositol (MI), an essential physiological metabolite, is among the key osmotically active compounds found in all eukaryotic cells [19]. Myo-inositol plays an important role not only in signal transduction, membrane biogenesis, cell wall formation, and phosphate storage, but also in the alleviation of abiotic stresses [20,21,22]. MI can combine with indole acetic acid (IAA) to form IAA-MI, which promotes the long-distance transport of endogenous auxin in plants [23]. The exogenous application of MI regulates plant transpiration water loss and carbon dioxide absorption for photosynthesis by modifying stomatal aperture [24]. It has been shown that MI is also involved in ion toxicity and osmotic balance regulation, and that it alleviates salt stress by reducing Na^+^ accumulation and sodium potassium ratios and partially regulating the accumulation of osmotic substances such as soluble sugars [25], as supported by the promoted wheat growth under salt stress after exogenous application of MI [26]. Another example is that MI increased the photosynthetic rate by maintaining a higher chlorophyll content in maize seedlings under salt stress [27]. Furthermore, MI significantly reduced the accumulation of hydrogen peroxide and superoxide in plants, mitigated lipid peroxidation and electrolyte leakage, and enhanced cell membrane stability [25,28].

To investigate the growth-promotion and stress-tolerance effects of MI and waste-derived amendments on vegetable crops, our study focused on the effects of varied concentrations of MI and CSL co-application on the growth and stress resistance of cabbage seedlings. Our goal is to identify the optimal concentrations and ratios to maximize the biomass of cabbage, providing technical guidance for promoting early crop growth and salt tolerance.

## 2. Results

### 2.1. Analysis of the Effects of CSL, MI, and Their Combination on Cabbage Growth and Salt Tolerance

Compared to the normal growth condition, the addition of 150 mM NaCl greatly inhibited the growth of Chinese cabbage seedlings, as evidenced by the sharp decrease in the aboveground fresh weight (Figure 1C). However, when different concentrations of CSL, MI, or CSL-MI complex were applied under salt stress, a remarkable increase in the aboveground fresh weight of cabbage was observed. Among them, the effects of the C1M3 (0.1 mL L^−1^ CSL + 288 mg L^−1^ MI) treatment and the C2M3 (0.6 mL L^−1^ CSL + 288 mg L^−1^ MI) treatment were the most pronounced, increasing by 89.80% and 85.71%, respectively (Figure 1C). Similar trends were also observed in the aboveground dry weight of cabbages under salt stress with varying nutrient addition (Figure 2C). Salt stress also significantly inhibited the plant height of cabbage, resulting in a 29.36% decrease compared to the control. However, the application of CSL, MI, and the CSL-MI combination rescued the growth of cabbage inhibited by salt stress. Among these treatments, C1M3 showed the most significant promotion effect, with a 37.39% increase compared to the N treatment. Therefore, the application of CSL, MI, and their combination significantly improved their growth performance, with the most pronounced effects observed when MI and CSL-MI combination was provided (Figure 1B,D).

Reactive oxygen species (ROS), such as hydrogen peroxide (H_2_O_2_) and superoxide (O_2_^−^), play a crucial role in stress responses. In this study, we visualized H_2_O_2_ and superoxide O_2_^−^ accumulation in cabbage leaves under salt stress by in situ staining with 3,3′-diaminobenzidine (DAB) and nitro blue tetrazolium (NBT), respectively. Compared to the control, cabbage leaves under salt stress conditions exhibited intensified dark brown (staining by DAB) and deep blue colors (staining by NBT), with staining intensity positively correlating with the severity of stress experienced by the cabbage (Figure 1B,E,F). In contrast to the salt stress treatment, the application of MI, CSL, and their combination significantly reduced the production of H_2_O_2_ and superoxide in cabbage seedlings under salt stress (Figure 1B,E,F).

### 2.2. Analysis of the Effects of CSL, MI, and Their Combination on Cabbage Leaf Photosynthetic Pigments

Compared to the control, the NaCl treatment significantly reduced the contents of photosynthetic pigments, including chlorophyll a, chlorophyll b, and carotenoids, in cabbage (Figure 2). However, the application of CSL or MI alone, as well as their combination, mitigated this decrease to varying extents. Compared to the NaCl treatment, the C1 (0.1 mL L^−1^ CSL), but not C2 (0.6 mL L^−1^ CSL) treatment, significantly increased chlorophyll a, b, and carotenoids contents in cabbages (Figure 2A,B,D). Similarly, compared to NaCl treatment, the M2 treatment (72 mg L^−1^ MI) increased chlorophyll a, and both M1 (18 mg L^−1^ MI) and M2 treatments increased chlorophyll b and carotenoid contents significantly under salt stress. The combination of MI and CSL outperformed individual CSL or MI treatments in enhancing photosynthetic pigment content in cabbage. The C1M1, C1M2, and C1M3 treatments showed the most pronounced promotion effects on chlorophyll a content in cabbage, increasing by 54.84%, 43.55%, and 46.77%, respectively. In addition, the C1M1 and C1M3 treatments significantly increased the content of chlorophyll b in cabbage by 37.84% and 32.43%, respectively. Finally, the C1M1, C1M2, C1M3, and C2M2 treatments showed the most pronounced improvement in carotenoid content.

### 2.3. Analysis of the Effects of CSL, MI, and Their Combination on Physiological Characteristics of Cabbage Leaves under Salt Stress

#### 2.3.1. Malondialdehyde (MDA) Content

Compared to the control treatment, the MDA content in cabbage significantly increased under salt stress (NaCl treatment), with an increase of 83.58%, indicating MDA as a biomarker of stress stimulation in plants (Figure 3A). However, the application of CSL, MI, or their combination significantly reduced the MDA content in cabbage to a nearly normal level. Notably, there were no significant differences in the reduction in MDA content among individual CSL, MI, or their combination, except that the effect of M3 (288 mg L^−1^ MI) treatment was slightly weaker than other treatments (Figure 3A).

#### 2.3.2. Electrolyte Leakage (EL)

The application of 150 mM NaCl significantly increased the electrolyte leakage permeability of the cabbage cell membrane (Figure 3B), and subsequent CSL treatment did not reverse this change. On the contrary, the M1 and M2 treatments, but not the M3 treatment, significantly decreased the REP of cabbage by 21.93% and 23.26%, respectively, compared to the NaCl treatment. In addition, although the C1M1 and C2M1 treatments did not show a significant influence, the C1M2, C1M3, C2M2, and C2M3 treatments significantly reduced the REP of cabbage by 20.93%, 23.08%, 12.75%, and 15.90%, respectively.

### 2.4. Analysis of Changes in the Content of Soluble Sugars and Soluble Proteins

Salt stress (NaCl treatment) significantly reduced the content of soluble sugars in cabbage (Figure 3C). However, treatment with CSL, MI, and their combination prevented plants from this decrease except for the 0.1 mL L^−1^ CSL treatment. Particularly, the M2 treatment displayed the most effective promotion, increasing soluble sugar content by 153.22% compared to the NaCl treatment. The combination treatments C1M3 and C2M3 also demonstrated notable efficacy, promoting soluble sugar increase by 95.22% and 95.91%, respectively. Comparatively, the CSL treatment significantly influenced the content of soluble proteins in cabbage under salt stress (Figure 3D). In the MI treatments, M2 showed the most pronounced impact on soluble protein contents. The combinations of MI and CSL materials also significantly increased the content of soluble proteins under salt stress, with the C1M3 treatment showing the most substantial effect.

### 2.5. Analysis of Ionic Homeostasis Changes

#### 2.5.1. Changes in Na^+^, K^+^, Ca^2+^, and Mg^2+^ Ion Contents

Compared to the control, the application of 150 mM NaCl significantly increased the Na^+^ content and decreased the K^+^, Ca^2+^, and Mg^2+^ contents in cabbage (Figure 4). Under salt stress, the application of MI significantly reduced the Na^+^ content in cabbage, with the M2 treatment showing the most significant effect. In the combination treatment of MI and CSL, the C2M1 and C2M2 treatments significantly increased the Na^+^ content, while the C1M1, C1M3, and C2M3 treatments significantly reduced the Na^+^ content (Figure 4A). Compared to the N treatment, the C1M3 treatment significantly increased the K^+^ content in cabbage by 42.37% (Figure 4B). Application of the stress protectants under salt stress significantly promoted the elevation of Ca^2+^ content in cabbage, with the C1, C1M3, C2M1, C2M2, and C2M3 treatments demonstrating the most significant effects (Figure 4C). Similarly, under salt stress, application of CSL and MI significantly promoted the increase in Mg^2+^ content in cabbage (Figure 4D). Among treatments where MI and CSL were applied separately, the C1 treatment had the most significant effect on Mg^2+^ content changes. After the combination of MI and CSL, the C1M3, C2M1, C2M2, and C2M3 treatments showed the best promotion effects.

#### 2.5.2. Analysis of Changes in Ion Ratios

Compared to the control, the application of 150 mM NaCl significantly increased the sodium/potassium ratio (Figure 5A). However, the application of enhancers significantly reduced the sodium/potassium ratio, with the C1M3 treatment showing the most significant effect, followed by the C2M3 treatment, resulting in reductions of 40.48% and 32.12%, respectively. Salt stress significantly increased the sodium/calcium ratio and sodium/magnesium ratio in cabbage (Figure 5B,C). Compared to the NaCl treatment, the C1M3 treatment showed the most effective reduction, with decreases of 44.39% and 43.21% in the sodium/calcium ratio and sodium/magnesium ratio, respectively. The C2M3 treatment showed the second-best effect, resulting in reductions of 38.09% and 38.92% in the sodium/calcium ratio and sodium/magnesium ratio, respectively.

### 2.6. Principal Component and Correlation Analysis of the Effects on the Growth and Physiological Characteristics of Cabbage

#### 2.6.1. Principal Component Analysis

Principal component analysis of cabbage indicators revealed a significant difference between the N treatment and other treatments (Figure 6A,B). Analyzing the first principal component (PC1) demonstrated that the combination treatments of MI and CSL, including the C1M2, C1M3, and C2M3 treatments, significantly increased the aboveground fresh weight, plant height, and photosynthetic pigments, while decreasing the MDA, REP, and Na^+^ contents under salt stress. Analysis of the second principal component (PC2) indicated that, compared to separate applications of MI and CSL, the combination treatment of MI and CSL substantially increased the K^+^, Ca^2+^, and Mg^2+^ contents in cabbage. Among the combination treatments of MI and CSL, the C1M3 treatment exhibited a particularly significant effect.

#### 2.6.2. Correlation Analysis

As shown in Figure 6C, the aboveground fresh weight of cabbage exhibited significant positive correlations with the aboveground dry weight, plant height, chlorophyll a, chlorophyll b, carotenoids, soluble proteins, and soluble sugars. In contrast, it was significantly negatively correlated with MDA, EL, and Na^+^. The aboveground dry weight of cabbage displayed a strong positive correlation with plant height, carotenoids, and soluble sugars while maintaining a strong negative correlation with EL and Na^+^. Furthermore, the plant height of cabbage was highly positively correlated with chlorophyll a, chlorophyll b, carotenoids, soluble proteins, and soluble sugars, but negatively correlated with MDA, EL, and Na^+^ (Figure 6C and Appendix A).

## 3. Discussion

Salt stress poses a significant threat to global food security. Enhancing plant tolerance through the application of protective agents is considered an effective strategy to mitigate the impact of salt stress on plant growth. The present study unveils the beneficial effects of CSL, MI, and CSL-MI when applied under salt stress conditions on the growth and stress tolerance of seedling cabbage. Salt stress affects plant growth by inducing osmotic and ionic stress. Additionally, it restricts plant growth by manifesting in behaviors such as limiting photosynthesis, decreasing enzyme activity, and disrupting normal metabolism [29,30].

CSL is rich in nutrients essential for crop growth and development, exerting a growth-promoting and yield-increasing effect [31]. MI, recognized as a growth factor, actively participates in the physiological and metabolic processes of plants, promoting overall plant growth [21]. In this study, the application of CSL, MI, and the combination of CSL-MI all contributed to the increased biomass of cabbage under salt stress conditions, with the treatment of 0.1 mL L^−1^ CSL + 288 mg L^−1^ MI showing the most significant effect. This suggests that the combined application of CSL and MI has a superior enhancing effect on cabbage growth compared to their individual application.

Photosynthesis, a crucial physiological process in plants, can be negatively impacted by salt stress, hindering plant growth and development [32]. The decrease in chlorophyll levels is considered as a typical symptom of oxidative stress in plants after exposure to salt stress [33]. The primary reasons for this phenomenon are the inhibition of chlorophyll synthesis and the activation of chlorophyll-degrading enzymes [34]. Additionally, carotenoids, functioning as antioxidants, may protect plants from the effects of reactive oxygen species [35]. Navarro-Morillo et al. demonstrated that the exogenous root application of CSL alleviated the adverse impact of salt stress on plant photosynthesis [11]. Similarly, MI enhanced photosynthesis in plants by increasing the content of photosynthetic pigments and stomatal conductivity [25]. In this study, the application of CSL also alleviated the detrimental effects of salt stress cabbage photosynthesis, while exogenous MI application increased the content of photosynthetic pigments in cabbage. The combined application of MI and CSL increased the photosynthetic pigment content more effectively than the separate CSL and MI applications, indicating a synergistic effect between the two materials.

Under salt stress conditions, cabbage experienced an overaccumulation of ROS, resulting in cellular oxidative damage to cellular components [36]. Our study demonstrated that the application of myo-inositol, corn steep liquor, and their combination materials significantly reduced the ROS levels in the cabbage under salt stress conditions. MDA is produced from cell membrane lipid peroxidation and often serves as an indicator of salt stress-induced damage to plant cells [37]. Under salt stress, MDA content typically increases in plants. In this study, the application of CSL, MI, and the combination material of CSL-MI significantly reduced the MDA content in cabbage. This aligns with findings by Navarro-Morillo et al. that the root application of CSL effectively reduced the MDA content in pepper plants [11]. Additionally, Al-Mushhin et al. reported that exogenous MI application reduced MDA content and alleviated oxidative damage caused by salt stress to quinoa [38].

The cell membrane is one of the primary sites of damage in plants. EL is an important parameter for assessing cell membrane permeability in plants exposed to stress, and alterations in EL correlate with the degree of crop damage under adverse conditions. As demonstrated by Hu et al., the exogenous application of MI alleviated the impact of salt stress on cell membrane permeability in plants. In this study, the application of 18 and 72 mg L^−1^ MI significantly reduced the leaf EL of cabbage under salt stress conditions, while the effect of CSL on EL was not obvious. Notably, the combined application of CSL and MI, specifically the treatments of 0.1 mL L^−1^ CSL + 72 mg L^−1^ MI and 0.1 mL L^−1^ CSL + 288 mg L^−1^ MI significantly reduced the leaf EL of cabbage, emphasizing the primary role of MI in mitigating EL.

Soluble sugars and soluble proteins function as osmoprotectants in plant cells to maintain osmotic balance [39]. Our experiment showed a significant decrease in the content of soluble sugars and soluble proteins under salt stress conditions. Application of CSL, MI, and combination materials increased the content of both in cabbage, with the treatments of 72 mg L^−1^ MI and 0.1 mL L^−1^ CSL + 288 mg L^−1^ MI showing the most effective improvement. This occurred because CSL and MI regulate the accumulation of some osmolytes in plants to counteract the effects of salt stress [11,25].

Salt stress can cause ion toxicity in plants, disrupting ion homeostasis and affecting the physiological metabolic processes, ultimately affecting plant growth and development [40]. Salt stress alters the absorption, transportation, and distribution of major ions such as K^+^, Na^+^, Ca^2+^, and Mg^2+^ in plants. In this study, under 150 mM NaCl stress, the Na^+^ ion content in cabbage increased significantly. Excessive Na^+^ in cabbage competed with other cations and diminished the absorption of K^+^, Ca^2+^, and Mg^2+^. Concurrently, the Na^+^/K^+^, Na^+^/Ca^2+^, and Na^+^/Mg^2+^ values in cabbage also greatly decreased, indicating that 150 mM NaCl stress disrupted the ion balance in cabbage. Our findings in this study showed that although CSL was nutrient-rich, its high salt content elevated the Na^+^ levels in plants and also increased the K^+^, Ca^2+^, and Mg^2+^ contents, resulting in a decrease in Na^+^/K^+^, Na^+^/Ca^2+^, and Na^+^/Mg^2+^ values in cabbage. This suggests that the exogenous application of CSL can alleviate the impact of salt stress on cabbage ion balance. Similar ion changes also occurred in cabbages after exogenous application of MI, which is consistent with the results obtained by Hu et al. on MI application to apple, supporting MI’s efficacy in maintaining the ion balance in crops under salt stress conditions [11]. Remarkably, our findings revealed treatments of 0.1 mL L^−1^ CSL + 288 mg L^−1^ MI and 0.6 mL L^−1^ CSL + 288 mg L^−1^ MI outperformed CSL or MI application alone, demonstrating a synergistic effect of CSL and MI in regulating plant ion balance under salt stress.

Principal component analysis revealed that the combined application of CSL and MI showed stronger correlations across various indicators compared to their individual application. Especially, the treatment of 0.1 mL L^−1^ CSL + 288 mg L^−1^ MI showed the strongest correlation with various indicators. Correlation analysis between cabbage biomass and functional traits found that cabbage growth was primarily related to photosynthetic pigment content, soluble sugars, soluble proteins, MDA, EL, and Na^+^, indicating that the growth of cabbage is primarily related to and influenced by these indicators. The notable decrease in MDA levels suggests that the treatments involving MI and CSL effectively counteracted the oxidative damage induced by NaCl [11,38]. Conversely, the relatively minor impact on electrolyte leakage may imply that, while oxidative stress was mitigated, there might still be some compromise to cellular membrane integrity [25]. Thus, it is worth noting that the reduction in Na^+^ content may not be the primary mechanism through which CSL and MI exert their protective effects. Alternatively, the treatments might be influencing ion compartmentalization within specific cellular or subcellular compartments rather than directly affecting the overall cellular Na^+^ content. Further exploration is needed to delve into this aspect and gain a more comprehensive understanding of the mechanisms underlying the protective effects of MI and CSL.

In summary, the combined application of CSL and MI showcased significant promotion of Chinese cabbage seedling growth and stress resistance, mainly through increasing the photosynthetic pigments, maintaining cell membrane permeability and ion permeation balance, and reducing ROS generation (Figure 5D). The combination of MI and CSL enhances their individual benefits, leveraging CSL’s nutrient supply capability. This approach effectively utilizes industrial and agricultural waste and provides an economical strategy for enhancing crop growth and stress resistance. Furthermore, this study identified that an application concentration of 72 mg L^−1^ of MI proved to be optimal in the sole MI treatment. This finding aligns with the results reported by Hu et al., indicating an optimal application concentration for MI [25]. Notably, research suggests that varying levels of MI play a crucial role in regulating plant growth and development by mediating ethylene and brassinosteroids [41]. As a growth factor, MI’s impact on plant growth is multifaceted, encompassing its involvement in stress responses, signaling pathways, and physiological processes [42,43]. The intricate nature of MI’s influence on plant growth underscores the notion that higher concentrations of inositol may not always yield superior results. In fact, excessive concentrations could disrupt pathways, leading to imbalances and, consequently, suboptimal or inhibitory effects on plant growth.

## 4. Materials and Methods

### 4.1. Materials

The MI and CSL used in this experiment were obtained from Shandong Zhucheng Haochen Biotechnology Co., Ltd., Zhucheng, China. The MI was in the form of white powder. The CSL was a yellow-brown liquid, with the following chemical properties: pH 4.50, EC 49.90 mS·cm^−1^, total nitrogen 30.70 g·L^−1^, phosphorus (P_2_O_5_) 17.40 g·L^−1^, potassium (K_2_O) 52.40 g·L^−1^, micro element 0.11 g L^−1^, total organic carbon 75.70 g·L^−1^, organic acid 298 g·L^−1^, amino acid 54.70 g·L^−1^, and sugar 32.50 g·L^−1^.

### 4.2. Plant Growth Condition and Experimental Design

Cabbage (*Brassica rapa* subsp. *pekinensis*) “Jing Cui 60” cultivar was grown in the greenhouse of China Agricultural University from June to July 2023. The photoperiod of the greenhouse was 16 h of light/8 h of darkness, with an average relative humidity of 70–75%, a diurnal photosynthetically active radiation (PARE) of 750 ± 20 µmol m^−2^ s^−1^, and a temperature of 28/15 °C.

Disinfected cabbage seeds were sown in nursery trays (length, width, and height: 28 cm × 28 cm × 6 cm, 25 holes per burrow tray) filled with substrate (Pindstrup, Danmark), which consisted of potassium 9.42 mg kg^−1^, phosphorus 190.42 mg kg^−1^, inorganic nitrogen 29.86 mg kg^−1^, organic matter 176.65 g kg^−1^, cation exchange capacity (CEC) 26.85 mol kg^−1^, and pH 5.5–6.6. Three-hole trays were set up for each treatment, and 25 cabbage plants were planted in each hole tray. When the cabbage reached the two-leaf-one-heart stage (day 18 after sowing), they were irrigated with water containing 150 mM NaCl at a rate of 250 mL/day per tray for 4 days. One day after the end of NaCl irrigation, the substrate received a one-time treatment with MI and CSL at a rate of 250 mL/tray, following the concentration gradient outlined in Figure 1A. The final salt content (NaCl) in the substrate was 8.775 g per tray. Soil moisture in the substrate was carefully regulated to maintain approximately 60% during the planting period, with consistent application of insect control and other management practices.

### 4.3. Determinations and Measurements

On the 25th day after sowing, when cabbage seedlings reached the 5-leaf stage, samples were collected for growth and physiological measurements. Four independent biological replicates from the first true leaf of the cabbage were conducted for each treatment and each replicate consisted of 9–12 plants.

The following vegetative parameters were measured: aboveground fresh weight, plant height (distance from the base of the plant to the tip of the highest leaf), and aboveground dry weight (obtained by drying the aboveground parts at 105 °C for 30 min, followed by 65 °C for 48 h before weighing).

Photosynthetic pigment content: 10 discs per replicate (each replicate pool including 9–12 individual plants) from the first true leaf of the cabbage, that is, in total, four replicate pools (40 discs in total), were sampled for treatment. A total of 40 discs, each with a diameter of 1 cm, were punched from the first true leaf. Chlorophyll was extracted in 15 mL of 95% ethanol. After 24 h in the dark, absorbance (649 nm, 665 nm, 470 nm) was measured to calculate pigment content following the method by Li [44].

Electrolyte leakage (EL) of leaves: The method used was modified from Dionisio-Sese and Tobita [45]. A total of 10 discs per replicate (each replicate pool including 9–12 individual plants) from the first true leaf of the cabbage, that is, in total, four replicate pools (40 discs in total), were sampled for each treatment. Each of the 40 discs of the first true leaf were submerged in a centrifuge tube containing 15 mL of distilled water. After shaking at 200 rpm at room temperature for 2 h, conductivity of the leaf solution was measured. The same leaf solution was then boiled at 100 °C for 10 min, cooled to room temperature, and shaken for an additional 2 h. The conductivity of the leaf solution was measured again, and relative electrolyte leakage was calculated.

Malondialdehyde (MDA): Four replicates (each replicate pool including 9–12 individual plants) from the first true leaf of the cabbage were sampled for each treatment. A total of 0.1 g of leaves were ground with 10% trichloroacetic acid (TCA) and centrifuged. Then, 2 mL of the supernatant was mixed with 2 mL of 0.6% thiobarbituric acid (TBA) solution. After 15 min of reaction in boiling water and rapid cooling, it was centrifuged for 10 min at 4000 rpm. The absorbance (532 nm, 600 nm, and 450 nm wavelengths) was measured to determine the malondialdehyde (MDA) content, referring to the method by Wang et al. [46].

Soluble sugars: Four replicates (each replicate pool including 9–12 individual plants) from the first true leaf of the cabbage were sampled for each treatment. A total of 50 mg of dried and ground cabbage leaves were repeatedly extracted with 80% ethanol. The supernatant obtained after centrifugation was decolorized with activated carbon for 30 min. The resulting solution was diluted to 10 mL with 80% ethanol, filtered, and the absorbance at 625 nm was measured to determine the soluble sugar content [46].

Soluble protein: 0.5 g of cabbage samples were ground to a homogenate in 5 mL of distilled water, and 0.1 mL of the supernatant, obtained after centrifugation, was mixed with 5 mL of Coomassie Brilliant Blue G-250 reagent. After 2 min of standing, the absorbance was measured at 595 nm to determine the soluble protein content [46].

Content of potassium, sodium, calcium, and magnesium ions: Determined by acid digestion-ICP-AES [47]. Four replicates (each replicate pool including 9–12 individual plants) from the first true leaf of the cabbage were sampled for each treatment. A total of 0.5 g of dried plant samples were ashed at 550 °C for 4 h in a muffle furnace. After cooling, the samples were dissolved with HCl solution, made up to volume, and filtered. The filtrate was measured using inductively coupled plasma atomic emission spectrometry (ICP-AES).

Histochemical detection of hydrogen peroxide (H_2_O_2_) and superoxide (O_2_^−^): According to El-Badri et al. [48], H_2_O_2_ and O_2_^−^ were localized by in situ staining using 3,3′-diaminobenzidine (DAB) and nitro blue tetrazolium (NBT), respectively. Cabbage leaves including four replicates (each replicate pool including 12 individual first true leaves from 12 individual plants) per treatment were stained with DAB and NBT solutions, and the images were quantified using ImageJ software (Version 1.54g, 2023).

### 4.4. Principal Component Analysis (PCA) and Correlation Analysis

To determine the relationship between the different treatments and each indicator, and the relationship between cabbage biomass and other indicators, PCA and correlation analysis were performed with IBM SPSS Statistics 26 and Origin 2023 software. By dividing the value of the component matrix corresponding to the principal component by the square root of the corresponding eigenvalue, the corresponding coefficient of each index in the two principal components was obtained; i.e., the coefficient = the value of the component matrix/sqtr (eigenvalue).

The salinity resistance indices were marked successively as X1, X2, X3, X4, and X5, and the standardized data were designated as ZXi. The obtained coefficient was multiplied by the corresponding standardized data to obtain the main component score:F1 = 0.278 × ZX1 + 0.268 × ZX2 + 0.221 × ZX3 + 0.219 × ZX4 − 0.235 × ZX5.(1)
F2 = −0.159 × ZX1 + 0.358×ZX2 − 0.647×ZX3 + 0.672 × ZX4 + 0.238 × ZX5.(2)

The proportion of the corresponding eigenvalues of each principal component to the sum of the total eigenvalues of the extracted principal component was used as the weight with which to calculate the principal component synthesis model, and the following synthesis model was obtained:F = 0.183 × ZX1 + 0.288 × ZX2 + 0.032 × ZX3+ 0.318 × ZX4 − 0.132 × ZX5.(3)

### 4.5. Data Processing and Statistics

IBM SPSS Statistics 26 was used for statistical analysis, employing one-way analysis of variance (ANOVA) with a significance level set at α = 0.05.

## 5. Conclusions

The root application of different proportions of CSL and MI effectively promoted the growth of cabbage seedlings under salt stress conditions. The exogenous application of CSL, MI, and the combination material of CSL-MI enhanced the photosynthesis of cabbage, protected cell membranes, and maintained cell osmotic and ion balance. Especially, the combined application of 0.1 mL L^−1^ CSL and 288 mg L^−1^ MI had the best effect.

## Figures and Tables

**Figure 1 plants-12-04110-f001:**
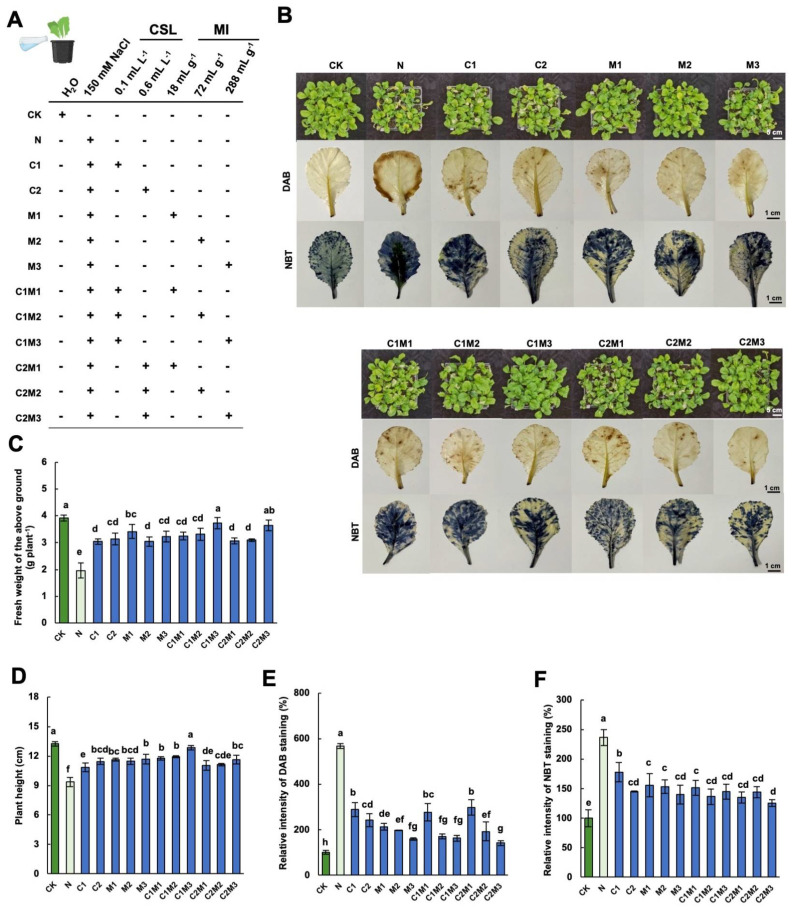
Analysis of CSL, MI, and their combination on the phenotypic and physiological changes in cabbage under salt stress. (**A**) Schematic diagram of experimental treatments. Effect of different concentration of MI, CSL, and the formulations on (**B**) phenotype, DAB (3,3′−diaminobenzidine), and NBT (nitro blue tetrazolium) staining of leaves, (**C**) fresh weight of aboveground, (**D**) plant height, (**E**) relative intensity of DAB staining, and (**F**) NBT staining comparing with control (CK) of cabbage under salt stress conditions. N, 150 mM NaCl treatment alone. The blue color on the leaves in (**B**) represents the degree of damage, with deeper staining in (**B**) indicating more severe damage. The data are shown as the means of four independent biological replicates, with each replicate pool consisting of 9–12 individual plants. Error bars represent standard deviations of three independent biological replicates. Lowercase letters on the graphs indicate significant differences (*p* < 0.05, Duncan’s test). Capital letters indicate different treatments which were listed in (**A**). Dark green columns are control treatments, light green columns are NaCl treatments, and blue columns are treatments with improved materials applied under salt stress conditions.

**Figure 2 plants-12-04110-f002:**
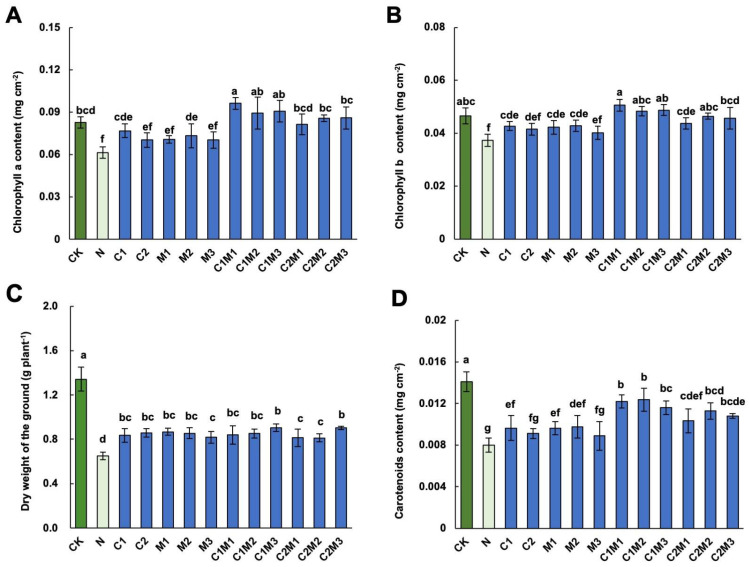
Analysis of CSL, MI, and their combination on cabbage leaf photosynthetic pigments under salt stress. Effect of CSL, MI, and their compound materials on the content of (**A**) chlorophyll a, (**B**) chlorophyll b, (**C**) dry weight, and (**D**) carotenoids of cabbage under salt stress. N, 150 mM NaCl treatment alone. The data are shown as the means of four independent biological replicates, with each replicate consisting of 9−12 individual plants. Error bars represent standard deviations of three independent biological replicates. Lowercase letters on the graphs indicate significant differences (*p* < 0.05, Duncan’s test). Capital letters indicate different treatments which were listed in Figure 1A. Dark green columns are control treatments, light green columns are NaCl treatments, and blue columns are treatments with improved materials applied under salt stress conditions.

**Figure 3 plants-12-04110-f003:**
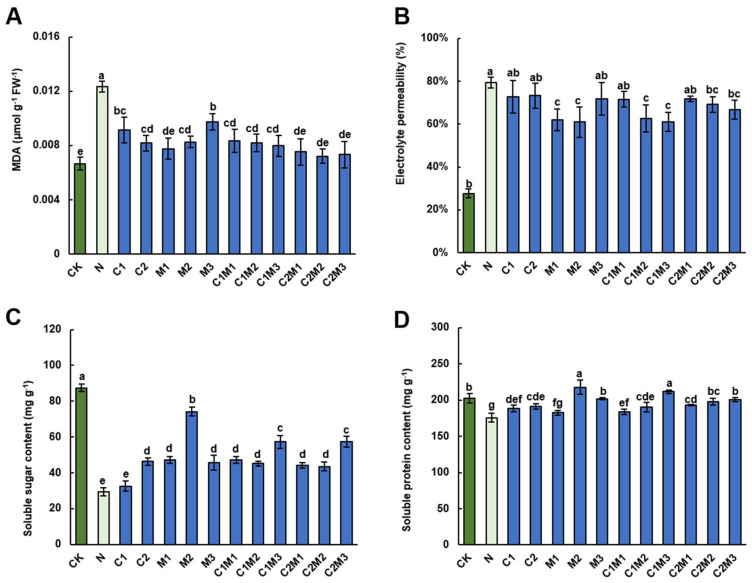
Analysis of CSL, MI, and their combination on salt-resistant physiological indicators of cabbage. Effect of CSL, MI, and their combination on (**A**) malondialdehyde (MDA) content, (**B**) electrolyte leakage (EL), (**C**) soluble sugar content, and (**D**) soluble protein content in cabbage under salt stress. The data are shown as the means of four independent biological replicates, with each replicate consisting of 9–12 individual plants. Error bars represent standard deviations of three independent biological replicates. Lowercase letters on the graphs indicate significant differences (*p* < 0.05, Duncan’s test). Capital letters indicate different treatments which were listed in Figure 1A. Dark green columns are control treatments, light green columns are NaCl treatments, and blue columns are treatments with improved materials applied under salt stress conditions.

**Figure 4 plants-12-04110-f004:**
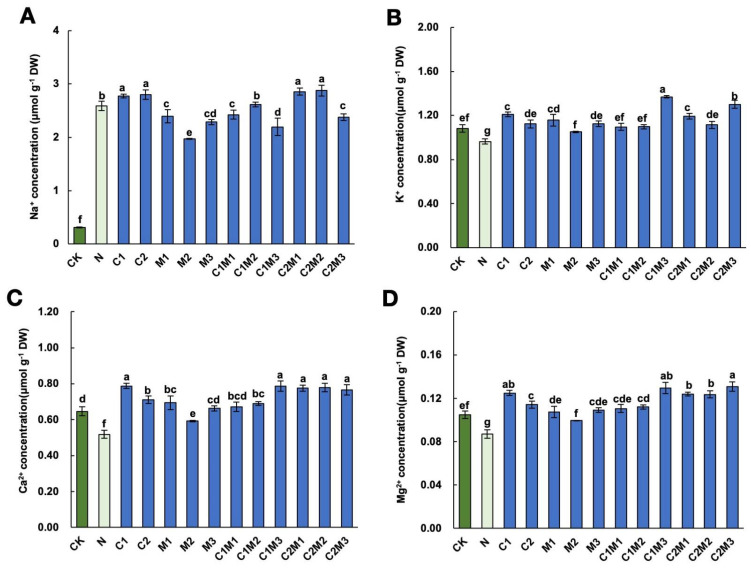
Effect of changes in sodium (Na^+^), potassium (K^+^), calcium (Ca^2+^), and magnesium (Mg^2+^) ion contents in cabbage under salt stress conditions by MI, CSL, and their combination. Effect of changes in (**A**) sodium (Na^+^), (**B**) potassium (K^+^), (**C**) calcium (Ca^2+^), and (**D**) magnesium (Mg^2+^) ion contents in cabbage under salt stress conditions by CSL, MI, and their combination. The data are shown as the means of four independent biological replicates, with each replicate consisting of 9–12 individual plants. Error bars represent standard deviations of three independent biological replicates. Lowercase letters on the graphs indicate significant differences (*p* < 0.05, Duncan’s test). Capital letters indicate different treatments which were listed in Figure 1A. Dark green columns are control treatments, light green columns are NaCl treatments, and blue columns are treatments with improved materials applied under salt stress conditions.

**Figure 5 plants-12-04110-f005:**
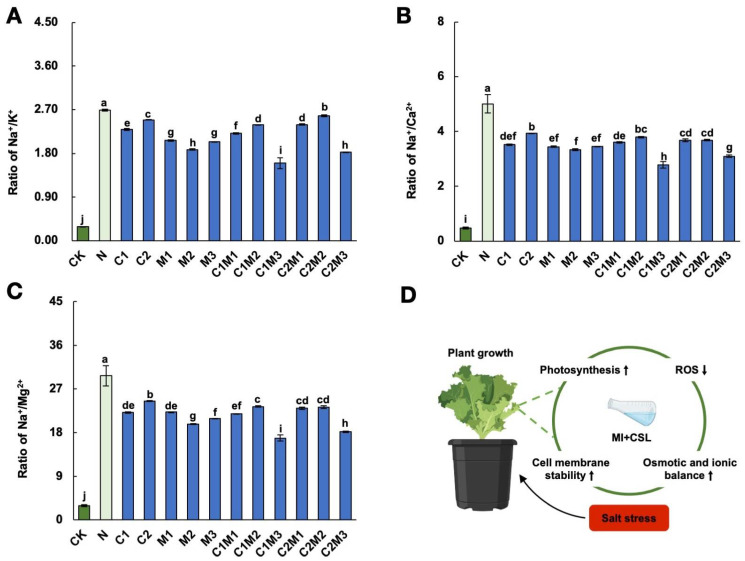
Effect of CSL, MI, and their combination on the ionic ratio of cabbage under salt stress. Effect of CSL, MI, and their combination on (**A**) Na^+^/K^+^, (**B**) Na^+^/Ca^2+^, and (**C**) Na^+^/Mg^2+^ of cabbage under salt stress. The data are shown as the means of three independent biological replicates, with each replicate consisting of 9–12 individual plants. Error bars represent standard deviations of three independent biological replicates. Lowercase letters on the graphs indicate significant differences (*p* < 0.05, Duncan’s test). Capital letters indicate different treatments which were listed in Figure 1A. Dark green columns are control treatments, light green columns are NaCl treatments, and blue columns are treatments with improved materials applied under salt stress conditions. (**D**) Schematic model by MI, CSL, and their combinations effect on the growth and salt stress responses of cabbage under salt stress.

**Figure 6 plants-12-04110-f006:**
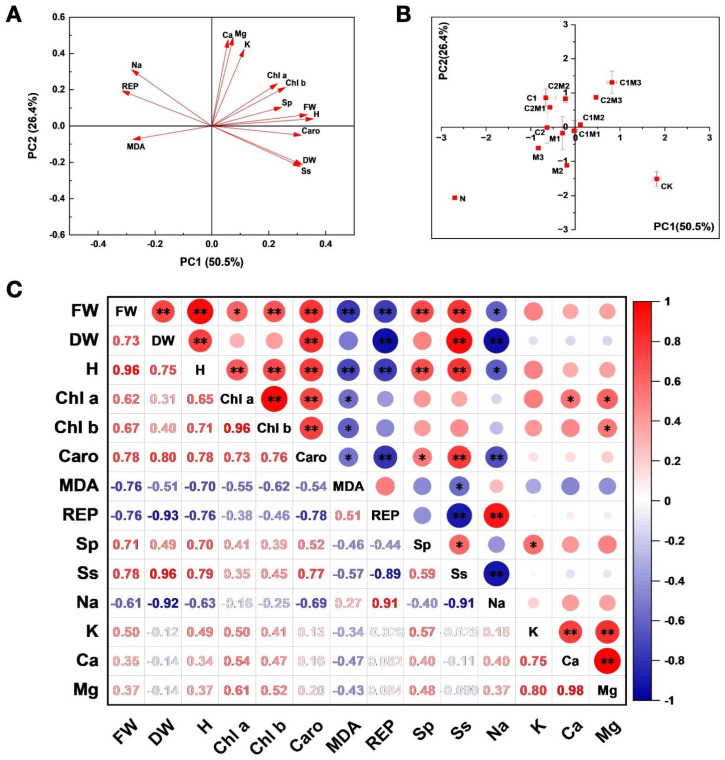
Analysis of the effects of CSL, MI, and their combination on the growth and physiological traits of cabbage under salt stress conditions. (**A**,**B**) Effect of MI, CSL, and their complexes on principal component analysis (PCA) of cabbage indicators under salt stress. The direction and length of the red arrows indicate the correlation. (**C**). Effect of MI, CSL, and their combination on correlations between all indicators of cabbage under salt stress. FW: fresh weight; DW: dry weight; H: plant height; Chl a: chlorophyll a; Chl b: chlorophyll b; Caro: carotenoid; Pro: proline; Sp: soluble protein; Ss: soluble sugar; REP: electrolyte leakage; MDA: malondialdehyde. The red−to−blue colored bar indicates the normalized relation strength between all indicators t of cabbage under salt stress. (** *p* < 0.01, * *p* < 0.05).

## Data Availability

All data generated in this study have been included in the article.

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
