# Peer review of "Combined Application of Myo-Inositol and Corn Steep Liquor from Agricultural Waste Alleviate Salt Stress in Brassica rapa"

_plants, 2023, doi:10.3390/plants12244110_

Round 1
Reviewer 1 Report
Comments and Suggestions for Authors
The manuscript describes the use of cornsteep liquor and myo-inositol to mitigate effects of salt stress in Chinese cabbage.
Several sentences in the abstract should be re-written to clarify meaning and more accurately reflect results. e.g
Salt stress poses a significant threat to crop production because it induces osmotic and ionic stress and disrupts nutrient absorption in plants.
'Different concentrations of myo-inositol and cornsteep liquor were applied to the roots, revealing that 150 mM NaCl significantly inhibited ..." This is not accurate. The fact that salt affected growth and physiology is separate to the mitigation of these effects by MI and CSL. These two matters should be clearly distinguished.
Page 2 paragraph 2 - 'The containing of bioactives...' - is poor English - replace with "The presence of bioactive molecules such as ..."
Avoid the use of 'can not only' - rewrite the sentences to avoid the use of the negative e.g. 'In summary, CSL can optimize soil structure, promote crop growth and development, and improve plant performance under abiotic stress.'
Page 2 paragraph 3 - '..belongs to one of the vital osmotic active substances,..' please rephrase.
Materials and Methods - please include information to clarify the relative timing of treatment application and tissue sampling. How many days after sowing were plants treated?
Clarify the replication for growth and physiological measurements. For example, in Figure 1 it is stated that there were 9-12 plants per replicate and 3 replicates per treatment. Are the data for each replicate pooled measurements of the 9-12 plants? Where leaf discs have been sampled how was this done? Was there 10 discs sampled per replicate i.e. 30 discs in total?
Figure 1A is incomplete and has errors, for example C1M3 contains M2. There are no rows for C2M1-3.
Page 6 - when describing treatment effects please clarify whether comparisons are being made with the CK or the N treatments.
Page 7 - it may be more appropriate to move the text describing DAB and NBT data to section 3.1.
Page 8 - refrain from using the term 'enhancing materials'. It may be better to say 'stress protectants' or something similar.
Page 11 - Figure 6C does not have data for proline but it is included in the caption. This Figure should be made larger to improve its readability. This could be done at the expense of Figure 6D which is not necessary.
Discussion - Be careful when describing effects of CSL and MI on plant growth and physiology to ensure that the context is correct i.e. with reference to salt-stressed plants. There is no data showing the effects of CSL and MI treatments in non-stressed plants. Be careful to go through the discussion carefully and remove any ambiguity. These treatments alleviate the effects of salt stress in Chinese cabbage but we do not know their effects on unstressed plants. Please also include some discussion on why M3 is often inferior to M2.
Comments on the Quality of English LanguageThe English is generally good although sentence structure and grammar could be improved. I have made some suggestions above.
Author Response
Comments and Suggestions for Authors
The manuscript describes the use of cornsteep liquor and myo-inositol to mitigate effects of salt stress in Chinese cabbage.
Several sentences in the abstract should be re-written to clarify meaning and more accurately reflect results. e.g Salt stress poses a significant threat to crop production because it induces osmotic and ionic stress and disrupts nutrient absorption in plants. " Different concentrations of myo-inositol and cornsteep liquor were applied to the roots, revealing that 150 mM NaCl significantly inhibited ..." This is not accurate. The fact that salt affected growth and physiology is separate to the mitigation of these effects by MI and CSL. These two matters should be clearly distinguished.
Answer: We have corrected it to a more appropriate description.
Page 2 paragraph 2 - 'The containing of bioactives...' - is poor English - replace with "The presence of bioactive molecules such as ..."
Answer: It has been corrected into the “The presence of bioactive molecules such as organic acids and plant growth regulators in CSL provides a basis for its use in soil improvement and biostimulant research”.
Avoid the use of 'can not only' - rewrite the sentences to avoid the use of the negative e.g. 'In summary, CSL can optimize soil structure, promote crop growth and development, and improve plant performance under abiotic stress.'
Answer: It has been corrected into the “In summary, CSL can optimize soil structure, promote crop growth and development, and improve plant performance under abiotic stress”.
Page 2 paragraph 3 - '..belongs to one of the vital osmotic active substances,..' please rephrase.
Answer: We have changed it to a more appropriate description.
Materials and Methods
- please include information to clarify the relative timing of treatment application and tissue sampling. How many days after sowing were plants treated?
Answer: Upon reaching the two-leaf stage (day 18 post-sowing), the cabbage plants underwent irrigation with water containing 150 mM NaCl at a rate of 250 mL/day per tray for a duration of 4 days. One day after NaCl irrigation, the substrate received a single treatment with Myo-inositol (MI) and Corn Steep Liquor (CSL) at a rate of 250 mL/tray, following the concentration gradient delineated in Fig. 1A. Soil moisture in the substrate was carefully regulated to maintain approximately 60% during the planting period, with consistent application of insect control and other management practices.
Clarify the replication for growth and physiological measurements. For example, in Figure 1 it is stated that there were 9-12 plants per replicate and 3 replicates per treatment. Are the data for each replicate pooled measurements of the 9-12 plants?
Answer: Yes, exactly. The detailed description has been added in the figure legends.
Where leaf discs have been sampled how was this done? Was there 10 discs sampled per replicate i.e. 30 discs in total?
Answer: 10 discs per replicate (each replicate pool including 9-12 individual plant) from the first true leaf of cabbage, that is, in total four replicate pools (40 discs in total), were sampled for each measurement. We have added the detailed sample replicate in the Materials and Methods part.
Figure 1A is incomplete and has errors, for example C1M3 contains M2. There are no rows for C2M1-3.
Answer: Corrected.
Page 6 - when describing treatment effects please clarify whether comparisons are being made with the CK or the N treatments.
Answer: We have added the appropriate description in the results part.
Page 7 - it may be more appropriate to move the text describing DAB and NBT data to section 3.1.
Answer: We have moved the text describing DAB and NBT data to section 3.1.
Page 8 - refrain from using the term 'enhancing materials'. It may be better to say 'stress protectants' or something similar.
Answer: It has been changed into the “stress protectants”.
Page 11 - Figure 6C does not have data for proline but it is included in the caption. This Figure should be made larger to improve its readability. This could be done at the expense of Figure 6D which is not necessary.
Answer: We have enlarged Figure 6C to make it visible easily, and moved Figure 6D into Figure 5D.
Discussion –
Be careful when describing effects of CSL and MI on plant growth and physiology to ensure that the context is correct i.e. with reference to salt-stressed plants. There is no data showing the effects of CSL and MI treatments in non-stressed plants. Be careful to go through the discussion carefully and remove any ambiguity. These treatments alleviate the effects of salt stress in Chinese cabbage but we do not know their effects on unstressed plants. Please also include some discussion on why M3 is often inferior to M2.
Answer: Thanks for your suggestions. We have improved the formulation in the discussion part. Indeed, lack of the data of CSL and MI effecting on non-stressed plants is a flaw in our experimental design, but for now we are focusing on the roles of CSL and MI on salt stress alleviation. We have added more discussion on why M3 is often inferior to M2 in the end of Discussion part.

Reviewer 2 Report
Comments and Suggestions for Authors
The paper is very interesting and the results can be very useful for scientists and growers dealing with NaCl salinity.
Many measurements are done to explain the effect of CSL and MI treatments at differnt doses and combinations in alleviation the salinity effects on cabbage seedlings. Please consider the number of treatments. On the Fig 1A there are only 10 and on the other figures 13. Fig 1A requires correction.
The parts; Results and Disscusion are correctly prepared and described, however the part Materials and Methods shoul be improved, especially the Figure 1A.
The corrections and suggestions are done in the text body.
I suggest to use in the text body and under the graphs the the right order od expressions: CSL and MI (alphabetical order)
Paper is worth to publish after some minor corrections.

Author Response
The paper is very interesting and the results can be very useful for scientists and growers dealing with NaCl salinity.
Many measurements are done to explain the effect of CSL and MI treatments at differnt doses and combinations in alleviation the salinity effects on cabbage seedlings. Please consider the number of treatments. On the Fig 1A there are only 10 and on the other figures 13. Fig 1A requires correction.
Answer: We have improved Figure 1A.
The parts; Results and Disscusion are correctly prepared and described, however the part Materials and Methods shoul be improved, especially the Figure 1A. The corrections and suggestions are done in the text body.
Answer: We have improved Materials and Methods section part in more detail, including greenhouse conditions, treatments, and biological replicates for samples.
I suggest to use in the text body and under the graphs the the right order od expressions: CSL and MI (alphabetical order)
Answer: We have corrected the order into right one: CSL and MI (alphabetical order).
Paper is worth to publish after some minor corrections.
Reviewer 3 Report
Comments and Suggestions for Authors
This is relatively interesting contribution to the field, but experimental design could have been better, including additional positive control with active products without salinity. Also, many experimental details are not known, not allowing for repetition of the experiment.
Title
I am afraid that it is not an "innovative approach". Better rephrase as "Combined application of myo-inositol and corn steep liquor from agricultural waste alleviate salt stress in Brassica rapa" or similar.
Abstract
Line 11–12, use "Salinity poses a significant threat to plant growth through induction of osmotic and ionic stress and disruption of nutrient absorption".
Use correct scientific name, Brassica rapa subsp. pekinensis.
Lines 16–17, start with describing the effect of salinity treatment. In the next sentence, write about additional treatment and move to describing the effects of that.
Lines 17 and 22, treatments were applied to substrate, not to roots.
Line 24, "the most pronounced positive effects".
Line 26, exclude "theoretical basis", as mechanistic aspects were not assessed or even properly discussed.
Keywords
"waste resource utilization" is not relevant.
Introduction
Line 34, reference to Ali et al. 2022 is not appropriate. These authors did not coin the term "physiological drought". Use some review paper defining these effects.
Lines 38–39, references Lin et al. 2012 and Panda & Khan 2009 are not appropriate for general description of negative effects of salinity, these are particular specific studies.
Line 42, use "napa cabbage" instead of "cabbage", use correct scientific name, Brassica rapa subsp. pekinensis.
I have doubts that Wu et al. 2021 (Research progress on the effects of salt stress on the growth and development of cabbage) is an appropriate reference on economic importance of napa cabbage. Similarly, did Yang et al. 2014 (Effects of NaCl stress on seed germination and seedling growth of cabbage) indeed defined that salt stress effects on cabbage cultivation affected the overall development of the vegetable economy?
Lines 52–53, it seems that it is not about "Agricultural waste" itself but about possibility to develop plant growth stimulators from it.
Lines 76–77, what is a reason to call myo-inositol as "crucial physiological metabolite"?
Lines 78–80, these two factual claims have no references.
Lines 84–85, this claim has no reference. In addition, "application" cannot "participate", it can lead to something.
The aim needs to be more precisely formulated, instead of describing experimental approach.
Materials and methods
It is very unfortunate that mineral nutrient composition of CSL is not provided.
Line 108, use "cultivar" instead of "variety".
In 2.2., many important details are missing, including light (PAR, photoperiod), temperature, humidity conditions in greenhouse, mineral nutrient availability in substrate, size of trays, number of plants per tray, watering regime (soil moisture), number of replicates, number of plants per replicate. Final concentration of sodium (salinity level) in experimental substrate is not known. As a result, it is not possible to repeat the experiment.
In 2.3., it is necessary to describe how representative sampling was performed, what was the number of biological and analytical replicates for each type of measurement. Not clear why the first true leaf was used for sampling in the case of pigments and electrolyte leakage, as these developed before any treatments have started. What material was used for other types of analysis?
There is no description of how DAB and NTB staining of leaves was performed and how quantitative results were obtained. How many leaves per treatment were used for that purpose to obtain comparable results instead of false (composite from individual leaves) visual comparison due to stochastic variation as shown in Fig. 1B?
As for experimental design, it is a significant shortage that no positive control (treatment with MI alone and CSL alone) has been included.
Performed PCA and correlation analysis are not even mentioned.
Results
In title to 3.1., use "combination" instead of "composition". Similarly in 3.2. and 3.3.
Instead of fresh weight of shoots and plant height, it is necessary to provide data on dry biomass of both shoots and roots separately.
Plant pictures in Fig. 1 are useless as cannot be used for comparison (copy-pasted single images without any size reference bars).
Line 171, use "most pronounced" instead of "most significant" to avoid confusion.
In the text, do not use abbreviation "CK" instead of "control", do not use abbreviation "N" instead of "salinity treatment" or "NaCl".
In the text, use "combination" instead of "composite".
Lines 204–205, 208–209, use "most pronounced" instead of "most significant".
In 3.3.2., instead of "relative electrolyte permeability", use "electrolyte leakage" as already used in Materials and methods. This concerns also Discussion section.
Lines 256–257, use "most pronounced" instead of "most significant".
Performed PCA analysis did not add much sense to the already seen and described effects. Similarly, correlation analysis was mostly useless, as the tightness had resulted from clearly asymmetric distribution of data from salt-treated plants as opposed to MI & CSL treated ones.
Discussion
Discussion is very limited and consists of short descriptions of single parameters. Broader picture is clearly missing, especially, related to scientific significance and novelty of the present study, as well as the limitations of the experimental system (young plants on the unknown background of mineral nutrient availability, with no additional positive control treatments (MI and CSL without salt) included.
There is no reason to tell about " This approach effectively utilizes industrial and agricultural waste and provides an economical strategy for enhancing crop growth and stress resistance", as these aspects were not assessed.
Some unanswered problems include but are not limited to the following. Why treatment with MI and CSL almost completely abolished NaCl-induced increase in MDA but their positive effects on electrolyte leakage were much less? Also, the actual degree of reduction in Na+ content by the treatments was relatively small and could not count for other effects.
Avoid using terms "root treatment" and "root application" as these are misleading.
Rather strange ethics approval note appears: "Purchase of all commercial pumpkin cultivars strictly followed both national and international guidelines".
List of references
Line 490, the reference to Liang et al. 2022 is incomplete and with erroneous title.
There are multiple other problems with the list that needs to be addressed.
Comments on the Quality of English LanguageModerate level language changes are necessary
Author Response
Answer to Reviewer #3
Comments and Suggestions for Authors
This is relatively interesting contribution to the field, but experimental design could have been better, including additional positive control with active products without salinity. Also, many experimental details are not known, not allowing for repetition of the experiment.
Title
I am afraid that it is not an "innovative approach". Better rephrase as "Combined application of myo-inositol and corn steep liquor from agricultural waste alleviate salt stress in Brassica rapa" or similar.
Answer: Corrected into “Combined application of myo-inositol and corn steep liquor from agricultural waste alleviate salt stress in Brassica rapa”.
Abstract
Line 11–12, use "Salinity poses a significant threat to plant growth through induction of osmotic and ionic stress and disruption of nutrient absorption".
Answer: Corrected into “Salinity poses a significant threat to plant growth through induction of osmotic and ionic stress and disruption of nutrient absorption”.
Use correct scientific name, Brassica rapa subsp. pekinensis.
Answer: Corrected.
Lines 16–17, start with describing the effect of salinity treatment. In the next sentence, write about additional treatment and move to describing the effects of that.
Answer: Corrected.
Lines 17 and 22, treatments were applied to substrate, not to roots.
Answer: Corrected into “substrate application”.
Line 24, "the most pronounced positive effects".
Answer: Corrected.
Line 26, exclude "theoretical basis", as mechanistic aspects were not assessed or even properly discussed.
Answer: “theoretical basis” has been excluded.
Keywords
"waste resource utilization" is not relevant.
Answer: “waste resource utilization” has been deleted.
Introduction
Line 34, reference to Ali et al. 2022 is not appropriate. These authors did not coin the term "physiological drought". Use some review paper defining these effects.
Answer: Corrected into other review articles.
Lines 38–39, references Lin et al. 2012 and Panda & Khan 2009 are not appropriate for general description of negative effects of salinity, these are particular specific studies.
Answer: Corrected.
Line 42, use "napa cabbage" instead of "cabbage", use correct scientific name, Brassica rapa subsp. pekinensis.
Answer: Corrected into “napa cabbage” and “Brassica rapa subsp. pekinensis.”
I have doubts that Wu et al. 2021 (Research progress on the effects of salt stress on the growth and development of cabbage) is an appropriate reference on economic importance of napa cabbage. Similarly, did Yang et al. 2014 (Effects of NaCl stress on seed germination and seedling growth of cabbage) indeed defined that salt stress effects on cabbage cultivation affected the overall development of the vegetable economy?
Answer: We have changed Wu et al. 2021 and Yang et al. 2014 into other references.
Lines 52–53, it seems that it is not about "Agricultural waste" itself but about possibility to develop plant growth stimulators from it.
Answer: We have improved the writing: “Agricultural wastes often contain bioactive substances, so the development of plant growth stimulants from agricultural wastes can play a positive role in promoting plant growth and resilience”.
Lines 76–77, what is a reason to call myo-inositol as "crucial physiological metabolite"?
Answer: The term "crucial physiological metabolite" is often used for myo-inositol due to its essential roles in various biological processes (Case et al., 2020). Myo-inositol serves as a precursor for important signaling molecules and is involved in cellular functions such as signal transduction, membrane integrity, and osmoregulation (Chen et al., 2010). Additionally, myo-inositol plays a vital role in the synthesis of phospholipids, which are crucial components of cell membranes. Its involvement in these fundamental processes makes it a crucial metabolite for the proper functioning of cells and overall physiological well-being.
Lines 78–80, these two factual claims have no references.
Answer: The references have been added.
Lines 84–85, this claim has no reference. In addition, "application" cannot "participate", it can lead to something.
Answer: We have improved the description of this section and included a reference citation.
The aim needs to be more precisely formulated, instead of describing experimental approach.
Answer: We have rephrased the goal accurately.
Materials and methods
It is very unfortunate that mineral nutrient composition of CSL is not provided.
Answer: We have added a more accurate mineral nutrient composition of CSL in Materials part.
Line 108, use "cultivar" instead of "variety".
Answer: Corrected into “cultivar”.
In 2.2., many important details are missing, including light (PAR, photoperiod), temperature, humidity conditions in greenhouse, mineral nutrient availability in substrate, size of trays, number of plants per tray, watering regime (soil moisture), number of replicates, number of plants per replicate. Final concentration of sodium (salinity level) in experimental substrate is not known. As a result, it is not possible to repeat the experiment.
Answer: We have added the details in the 2.2 Plant growth condition and Experimental Design.
The photoperiod of the greenhouse was 16 h of light/8 h of darkness, with an average relative humidity of 70-75%, a diurnal photosynthetically active radiation (PARE) of 750 ± 20 µmol m-2 s-1, and a temperature of 28/15 °C. Disinfected cabbage seeds were sown in nursery trays (length, width and height: 28 × 28 × 6 cm, 25 holes per burrow tray) filled with substrate soil (the main components were peat soil, woody peat, coco powder, grass peat and vermiculite, etc., pH 5.5-6.6, and ≥30% organic matter). Three-hole trays were set up for each treatment, and 25 cabbage plants were planted in each hole tray. When the cabbage reached the two-leaf-one-heart stage (day 18 after sowing), they were irrigated with water containing 150 mM NaCl at a rate of 250 mL/day per tray for 4 days. One day after the end of NaCl irrigation, the substrate received a one-time treatment with MI and CSL at a rate of 250 mL/tray, following the concentration gradient outlined in Fig. 1A. The moisture in the substrate soil was controlled to be maintained at about 60% during planting, and insect control and other management practices were kept the same. The final salt content (NaCl) in the substrate was 8.775 g per tray.
In 2.3., it is necessary to describe how representative sampling was performed, what was the number of biological and analytical replicates for each type of measurement. Not clear why the first true leaf was used for sampling in the case of pigments and electrolyte leakage, as these developed before any treatments have started. What material was used for other types of analysis?
Answer: We have added the sampling and the replicates detailed information in the 2.3 Plant growth condition and Experimental Design.
The first true leaf is representative of the first functional leaf of nape cabbage seedlings when plants subjected to salt stress. It has been supporting by previous reports in different crops suffering salinity stress, such as the first true leaf area and photosynthetic rates of cotton seedlings were positively associated with seedling vigor (Virk et al.,2020, Crop Science, 60(1), 404-418.). Furthermore, the first pair of true leaves of oil seed crop seedlings exhibited different adaptive strategies compared to the cotyledons under salt and alkali stresses, and the physiological traits of true leaves performed more sensitive responses to stress (Wang et al., 2019, Frontiers in plant science, 9, 1939.). To keep the consistency of the experiments, we always used the first or second true leaf for all analysis.
There is no description of how DAB and NTB staining of leaves was performed and how quantitative results were obtained. How many leaves per treatment were used for that purpose to obtain comparable results instead of false (composite from individual leaves) visual comparison due to stochastic variation as shown in Fig. 1B?
Answer: We have added more details in Material and methods- 2.3 Determinations and Measurements. Histochemical detection of hydrogen peroxide (H2O2) and superoxide (O2−) was performed according to (El-Badri et al, 2021), H2O2 and O2- were localised by in situ staining using 3,3′-diaminobenzidine (DAB) and nitroblue tetrazolium (NBT), respectively. Cabbage leaves including four replicates (each replicate pool including 12 individual first true leaves from 12 individual plants) per treatment were stained with DAB and NBT solutions, and the images were quantified using ImageJ software in Fig. 1B, 1E and 1F.
As for experimental design, it is a significant shortage that no positive control (treatment with MI alone and CSL alone) has been included.
Answer: In the previous studies, we have explored the effect of CSL and MI on the growth of cabbage seedlings under the normal growth condition. The results did not show any significant effects of CSL and MI (data not shown), also consider that the reports that MI mainly involved in the stress signal transduction (Kanter et al., 2005 Planta, 221, 243-254; Loewus et al.,2000 Plant science, 150(1), 1-19.; Mitsuhashi et al.,2008 Journal of experimental botany, 59(11), 3069-3076.), we only focused on the effect of salinity on cabbage seedlings in this study.
Performed PCA and correlation analysis are not even mentioned.
Answer: Principal Component Analysis and correlation analysis were completed in 2.4 Principal Component Analysis (PCA) and correlation analysis.
Results
In title to 3.1., use "combination" instead of "composition". Similarly in 3.2. and 3.3.
Answer: Corrected.
Instead of fresh weight of shoots and plant height, it is necessary to provide data on dry biomass of both shoots and roots separately.
Answer: The data for the dry weight of above-ground tissues were presented in Figure 2C. We did not harvest cabbage roots, because they were too slender and fragile to clean when they were cultivated in the substrates.
Plant pictures in Fig. 1 are useless as cannot be used for comparison (copy-pasted single images without any size reference bars).
Answer: The size of tray in Fig. 1 is 28 × 28 × 6 cm (length, width and height) consistently in all treatments. We used the size of tray as scar bars.
Line 171, use "most pronounced" instead of "most significant" to avoid confusion.
Answer: Corrected.
In the text, do not use abbreviation "CK" instead of "control", do not use abbreviation "N" instead of "salinity treatment" or "NaCl".
Answer: Corrected.
In the text, use "combination" instead of "composite".
Answer: Corrected.
Lines 204–205, 208–209, use "most pronounced" instead of "most significant".
Answer: Corrected.
In 3.3.2., instead of "relative electrolyte permeability", use "electrolyte leakage" as already used in Materials and methods. This concerns also Discussion section.
Answer: Corrected.
Lines 256–257, use "most pronounced" instead of "most significant".
Answer: Corrected.
Performed PCA analysis did not add much sense to the already seen and described effects. Similarly, correlation analysis was mostly useless, as the tightness had resulted from clearly asymmetric distribution of data from salt-treated plants as opposed to MI & CSL treated ones.
Answer: To avoid the asymmetric distribution of data from salt-treated plants, we did PCA again excluding the control and NaCl treatment (see Figure S1). The result shows the analysis of the effects of CSL, MI and their combinations on growth and physiological traits under salt stress conditions. The combination of MI and CSL was still strongly correlated with aboveground fresh weight, plant height and photosynthetic pigments positively, while highly negative correlated with MDA, EL and Na+.
Discussion
Discussion is very limited and consists of short descriptions of single parameters. Broader picture is clearly missing, especially, related to scientific significance and novelty of the present study, as well as the limitations of the experimental system (young plants on the unknown background of mineral nutrient availability, with no additional positive control treatments (MI and CSL without salt) included.
Answer: We have changed a few things in the first paragraph of discussion section.
There is no reason to tell about " This approach effectively utilizes industrial and agricultural waste and provides an economical strategy for enhancing crop growth and stress resistance", as these aspects were not assessed.
Answer: It has been rephrased.
Some unanswered problems include but are not limited to the following. Why treatment with MI and CSL almost completely abolished NaCl-induced increase in MDA but their positive effects on electrolyte leakage were much less? Also, the actual degree of reduction in Na+ content by the treatments was relatively small and could not count for other effects.
Answer: MDA serves as a marker for lipid peroxidation, serving as an indicator of oxidative stress. The notable decrease in MDA levels suggests that the treatments involving MI and CSL effectively counteracted the oxidative damage induced by NaCl. Conversely, the relatively minor impact on electrolyte leakage may imply that, while oxidative stress was mitigated, there might still be some compromise to cellular membrane integrity. Thus, it's worth noting that the reduction in Na+ content may not be the primary mechanism through which MI and CSL exert their protective effects. Alternatively, the treatments might be influencing ion compartmentalization within specific cellular or subcellular compartments rather than directly affecting the overall cellular Na+ content. Further exploration is needed to delve into this aspect and gain a more comprehensive understanding of the mechanisms underlying the protective effects of MI and CSL.
We have added this opinion in the discussion part.
Avoid using terms "root treatment" and "root application" as these are misleading.
Answer: Corrected.
Rather strange ethics approval note appears: "Purchase of all commercial pumpkin cultivars strictly followed both national and international guidelines".
Answer: Corrected.
List of references
Line 490, the reference to Liang et al. 2022 is incomplete and with erroneous title.
Answer: We have completed the reference.
There are multiple other problems with the list that needs to be addressed.
Answer: Corrected.
Round 2
Reviewer 3 Report
Comments and Suggestions for Authors
Thank you for corrections
Comments on the Quality of English LanguageOnly moderate editing is necessary